# HUDmax as a Novel Parameter in the Assessment of Ureteral Kinking: A Critical Evaluation for Predicting Ureteroscopic Lithotripsy Outcomes

**DOI:** 10.3390/medicina61091525

**Published:** 2025-08-25

**Authors:** Utku Can, Bilal Eryildirim, Alper Coşkun, Cengiz Çanakçı, Furkan Sendogan, Burak Doğrusever, Kemal Sarica

**Affiliations:** 1Department of Urology, Health Sciencies University, Kartal Dr. Lutfi Kirdar City Hospital, Istanbul 34865, Türkiye; bilaleryildirim@yahoo.com (B.E.); dr.alper05@gmail.com (A.C.); cengizcanakci@hotmail.com (C.Ç.); burakdogrusever@hotmail.com (B.D.); 2Department of Urology, Medicana Camlica Hospital, Istanbul 34692, Türkiye; furkandg@hotmail.com; 3Department of Urology, Sancaktepe Sehit Prof. Dr. Ilhan Varank Research and Training Hospital, Istanbul 34785, Türkiye; saricakemal@gmail.com; 4Department of Urology, Biruni University Medical School, Istanbul 34015, Türkiye

**Keywords:** ureteral kinking, ureteral stone, horizontal ureteral displacement, HUDmax, ureteroscopic lithotripsy

## Abstract

*Background and Objectives*: Ureteral kinking may hinder endoscopic access and reduce the success of ureteroscopic lithotripsy (URSL). This study evaluated whether kinking can be predicted preoperatively using non-contrast computed tomography (CT) by introducing a novel metric—Maximum Horizontal Ureteral Displacement (HUDmax)—and assessed its predictive value for procedural success. *Materials and Methods*: Data from 1261 patients who underwent URSL for a single ureteral stone were retrospectively analyzed. Patients were categorized into two groups based on whether the stone could be reached using a semirigid ureteroscope. Propensity score matching (1:2) was performed based on stone size and location, resulting in two matched cohorts: Group 1—Semirigid Inaccessible (SRI, *n* = 72), and Group 2—Semirigid Accessible (SRA, *n* = 144). Stone characteristics, ureteral wall thickness (UWT), and HUDmax were evaluated. Correlations between HUDmax and surgical parameters were analyzed, and the predictive value of HUDmax was assessed using receiver operating characteristic (ROC) analysis. *Results*: The SRI group showed significantly higher HUDmax values (median 2.36 mm vs. 1.2 mm, *p* < 0.0001). Semirigid access failure necessitated conversion to flexible ureteroscopy in all SRI cases, compared to 15% in the SRA group (*p* < 0.0001). Stone-free rates were significantly lower in the SRI group (45% vs. 82%, *p* < 0.0001), and the use of a double-J stent or nephrostomy placement was more frequent. Operative times were also longer in the SRI group (55 vs. 42 min, *p* < 0.0001). HUDmax correlated positively with operative time (*r* = 0.258, *p* = 0.005) but not with stone size, density, UWT, or hydronephrosis. ROC analysis showed HUDmax strongly predicted semirigid access failure (AUC: 0.805; cutoff: 1.58 mm), and moderately predicted stone-free status (AUC: 0.697; cutoff: 1.68 mm). *Conclusions*: Severe ureteral kinking constitutes a significant anatomical obstacle to the success of semirigid URSL. This study is the first to demonstrate that clinically relevant kinking can be predicted preoperatively using a non-contrast imaging modality, via the novel HUDmax parameter.

## 1. Introduction

The success of ureteroscopic lithotripsy (URSL) is influenced by several factors, including stone characteristics such as size, composition, and degree of impaction as well as the anatomy of the reno-ureteral unit [1]. Many parameters can be anticipated through the extensive use of preoperative computed tomography (CT) scans [2]. The location of the stone, in particular, is well-documented for its impact on procedural outcomes. For example, distal ureteral stones are typically associated with shorter operative times, lower complication rates, and higher success rates compared to proximal ureteral stones [3]. However, attributing these differences solely to the traversed distance oversimplifies the issue.

Although traditionally described as a straight structure within the retroperitoneum [4], the ureter often exhibits kinking, particularly in the proximal segment where it crosses the gonadal vein. This physiological bending, frequently accentuated during expiration, has been demonstrated through CT urography, retrograde pyelography, and cadaveric studies [5]. Another curve, not classified as a kink, is commonly observed at the iliac crossing level as the ureter enters the pelvis. Unlike the proximal ureter, this region is subject to extrinsic compression, representing one of the three physiological narrowing sites of the ureter. These anatomical features—kinking and extrinsic compression—have been suggested to complicate endourological procedures and contribute to ureteral stone obstruction [5]. However, the clinical significance of these variances, particularly kinking, remains largely unexplored in the literature.

The success of lithotripsy in ureteral stones often depends on navigating anatomical challenges, particularly in the presence of a narrow, angulated, or tortuous ureter. However, no current imaging modality reliably predicts such intraoperative difficulties. This study aimed to investigate whether ureteral kinking—an underrecognized anatomical factor that may impede endoscopic access—can be identified preoperatively using non-contrast computed tomography (NCCT). To achieve this, we introduced the Maximum Horizontal Ureteral Displacement (HUDmax), a novel, non-contrast CT-derived parameter designed to quantify the severity of kinking. The clinical relevance of HUDmax was evaluated by examining its association with semirigid access failure, operative time, and stone-free outcomes following URSL.

## 2. Materials and Methods

Departmental records of 3202 patients who underwent semirigid ureteroscopy (URS) for ureteral stones, with flexible URS additionally used in selected cases when needed based on intraoperative requirements, between 2014 and 2024 at our tertiary care center, were retrospectively reviewed. After applying exclusion criteria, 1261 patients with a single ureteral stone (5–15 mm in size) were included. Exclusion criteria included severe ureteral pathologies (e.g., ureteral or ureterovesical junction [UVJ] stenosis, ureteral polyps), preoperative ureteral stenting, renal-ureteral anomalies, significant skeletal deformities, and absence of non-contrast computed tomography (NCCT) within two weeks prior to surgery.

Subsequently, a 1:2 propensity score matching was performed based on stone size and location, resulting in a final cohort of 216 patients divided into two groups (see flow diagram in Figure 1):

Group 1: Semirigid Inaccessible (SRI), *n* = 72

Cases in which the stone could not be reached using a semirigid ureteroscope due to ureteroscopically confirmed kinking, tortuosity, or sharp angulation.

Group 2: Semirigid Accessible (SRA), *n* = 144

Cases in which the stone was fully visualized with a semirigid ureteroscope. This group also included cases with mild to moderate ureteral kinking, tortuosity, or angulation that were navigable with the semirigid ureteroscope.

Ureteral stones were classified based on standard anatomical landmarks: stones located below the sacroiliac joint were defined as distal ureteral stones, those between the sacroiliac joint and the lower border of the renal pelvis were classified as mid-ureteral stones, and those at or above the renal pelvis were considered proximal ureteral stones; only stones confined to these segments were included and categorized accordingly.

Demographic and clinical data were recorded, including age, gender, BMI, preoperative CT parameters (stone size, stone density, ureteral wall thickness [UWT], degree of hydronephrosis, and maximum horizontal ureteral displacement [HUDmax]), operative time, placement of a safety guidewire, need for double-J stent or nephrostomy placement, intraoperative complications, and hospital stay. Cases requiring transition to flexible ureteroscopy were also documented. Treatment success was defined based on stone-free status, which was defined as having residual fragments <3 mm, confirmed by X-ray for radiopaque stones and by NCCT for radiolucent stones within one week postoperatively.

### 2.1. Measurement Criteria

Stone size was calculated by multiplying the maximum diameter by the perpendicular diameter. This method allowed us to estimate the stone area [6]. Ureteral wall thickness was measured on the ureteral portion surrounding the stone at the point of longest soft-tissue thickness [7].

Maximum Horizontal Ureteral Displacement (HUDmax) (See Figure 2 and Figure 3): HUDmax quantifies the maximum horizontal displacement of the ureter distal to the stone using preoperative NCCT imaging in the axial plane. The calculation involves five steps:Tracing the Ureter: The ureter on the affected side is traced from the stone’s location to the iliac bifurcation.Identifying the location of maximum kinking: The region where the ureter exhibits the widest horizontal course is identified (indicated by arrows in Figure 3, Case 1a/2c).Measuring Horizontal Distance (y): The centers of the ureter at the CT slice where the kinking begins and the CT slice where it ends are identified. The horizontal distance between these two points is then measured (Figure 2 and Figure 3, Case 1b/2d).Measuring Vertical Height (h): Determined by multiplying the CT slice thickness by the number of slices spanning the kink (Figure 2). For example, for a slice interval of 2 mm over 4 slices, h = 2 × 4 = 8 mm.Calculating HUDmax: HUDmax is calculated by dividing the horizontal distance (y) by the vertical height (h): HUDmax = y/h.

In summary, HUDmax represents the horizontal displacement of the ureter per millimeter of vertical axis change, providing a quantitative measure of kinking severity.

### 2.2. Ureteroscopy Procedure

Under general anesthesia, all procedures were performed in the lithotomy position. A safety guidewire (Boston Scientific, Boston, MA, USA) was inserted at the beginning of each procedure. In cases where resistance was encountered, the guidewire was advanced under fluoroscopic guidance rather than using ureteroscopic visualization. The successful advancement of the guidewire beyond the proximal end of the stone was documented in all patients.

Semirigid ureteroscopes (6–7.5 Fr, Wolf, Knittlingen, Germany) were routinely employed. Flexible ureteroscopy (7.5 Fr, Karl Storz Flex X2, Karl Storz SE & Co. KG, Tuttlingen, Germany) was performed using a ureteral access sheath (9.5–11.5 Fr, Cook Flexor, Bloomington, IN, USA). Flexible ureteroscopy was required in all cases where the stone could not be reached with a semirigid ureteroscope, in cases of stone retropulsion into the kidney or when, based on the surgeon’s intraoperative judgment, flexible URS was considered preferable to optimize maneuverability, access difficult angles, or minimize ureteral trauma even if semirigid access was technically possible.

Stone disintegration was performed using a Holmium:YAG laser (365-µm fibers), and stone fragments were retrieved with forceps as necessary. Double-J stents were placed in cases with residual fragments or when intraoperative complications occurred, such as bleeding, ureteral perforation, or other suspected issues. Operative time was defined as the interval from the placement of the safety guidewire to the completion of the procedure.

This study protocol was approved by the Ethics Committee of Kartal Dr. Lütfi Kırdar City Hospital with Institutional Review Board (IRB) number 2023/514/260/7. The requirement for approval and written informed consent was waived by the committee.

### 2.3. Statistical Analysis

A one-to-two (1:2) propensity score matching was used to identify the nearest-neighbor matches between the groups, minimizing imbalance and confounding bias. Sample matching accounted for the following potential confounding variables: stone size and stone location. Categorical variables were compared using the chi-square test, while continuous variables were analyzed using Student’s *t*-test or Mann–Whitney U test. Bootstrapped Spearman correlation analysis (using 1000 samples and bias-corrected and accelerated confidence intervals) was employed to evaluate the relationships between HUDmax and continuous clinical variables. A 95% confidence interval excluding zero was considered indicative of a statistically significant correlation. ROC curve analysis evaluated HUDmax’s diagnostic performance. Statistical significance was set at *p* < 0.05, and analyses were conducted using Statistical Package for Social Sciences version 22.0 (SPSS, Chicago, IL, USA)

## 3. Results

### 3.1. Baseline Characteristics

Among the 216 patients, no significant differences were observed between the SRI and SRA groups in terms of age, gender, BMI, stone laterality, or degree of hydronephrosis. Ureteral wall thickness (UWT) was significantly lower in the SRI group (median 2.3 mm vs. 3.6 mm; *p* = 0.023). In the SRI group, 24% of stones were located in the mid-ureter and 76% in the proximal ureter, while no distal ureteral stones were identified. As propensity score matching was performed based on stone location, the SRA group exhibited the same distribution. Stone size and HU were comparable between the groups. HUDmax values were significantly higher in the SRI group (median 2.36 mm) compared to the SRA group (median 1.2 mm, *p* < 0.0001). In 74% of SRI cases, HUDmax was located within 2 cm distal to the stone, whereas in the SRA group, this rate was only 23% (*p* < 0.0001) (Table 1).

### 3.2. Surgical Outcomes

The median operative time was significantly longer in the SRI group (55 min vs. 42 min; *p* < 0.0001). All patients in the SRI group required conversion to flexible ureteroscopy after unsuccessful access with a semirigid ureteroscope. In contrast, 22 patients (15%) in the SRA group required conversion, primarily due to stone retropulsion (*n* = 18) or surgeon’s preference (*n* = 4) (*p* < 0.0001). Placement of a safety guidewire proximal to the stone at the beginning of the procedure was successful in 83% of SRI patients and 86% of SRA patients (*p* = 0.735). Stone-free status was achieved in 45% of the SRI group compared to 82% in the SRA group (*p* < 0.0001).

Following surgery, all patients in the SRI group required stenting: 94% received a double-J stent, and 6% (*n* = 4) required nephrostomy placement due to unsuccessful double-J stent insertion. In contrast, 83% of the SRA group received a double-J stent, while the remaining patients did not require any form of urinary drainage (*p* < 0.0001) (Table 1).

### 3.3. Correlations and Predictive Value of HUDmax

HUDmax demonstrated a significant positive correlation with operative time (r = 0.258; 95% CI, 0.089–0.433; *p* = 0.005). However, no significant correlations were found between HUDmax and stone size, density, UWT, or hydronephrosis grade (Table 2).

Receiver operating characteristic (ROC) analysis was conducted to evaluate the predictive utility of HUDmax for both semirigid ureteroscope accessibility (Figure 4A) and stone-free status (Figure 4B). In predicting failure of semirigid access, HUDmax yielded an area under the curve (AUC) of 0.805 (95% CI: 0.740–0.871), with an optimal cutoff value of 1.58 mm. This threshold provided 78% sensitivity and 75% specificity (*p* = 0.0001). In the separate analysis for predicting stone-free status, HUDmax demonstrated a moderate discriminative capacity with an AUC of 0.697 (95% CI: 0.620–0.773), and the optimal cutoff value was 1.68 mm, corresponding to 61% sensitivity and 71% specificity (*p* = 0.0001). These results indicate that while HUDmax is a strong predictor of semirigid access failure requiring flexible ureteroscopy, its ability to predict stone-free outcomes is comparatively lower.

## 4. Discussion

This study is the first to introduce a simple, non-contrast CT-based quantitative parameter, HUDmax, for evaluating ureteral kinking and its impact on the outcomes of semirigid ureteroscopic lithotripsy (URSL). While ureteral kinking has long been recognized as an anatomical variation that can complicate endoscopic procedures, its objective assessment and predictive value for URSL success have not been previously established. By quantifying the maximum horizontal displacement of the ureter, our results demonstrate that higher HUDmax values are significantly associated with semirigid access failure, longer operative times, and lower stone-free rates. These findings address an important unmet need for preoperative prediction of anatomically challenging URSL cases and complement previous anatomical classifications by providing a practical, reproducible measurement.

The development of HUDmax also builds on lessons learned from previous visual-based classification attempts. For instance, the system proposed by Kamo et al. [5] categorized upper ureteral kinking into three grades based on 3D CT urography findings, but its clinical significance for predicting URSL success or spontaneous stone passage remained unproven. One limitation was the reliance on contrast-enhanced imaging, which is not routinely performed preoperatively. To overcome this, our study utilized non-contrast computed tomography (NCCT)—a standard modality—to define HUDmax as a reliable, quantitative parameter. HUDmax demonstrated strong diagnostic performance for predicting semirigid ureteroscope access failure, with a cutoff value of 1.58 mm (AUC: 0.805 [95% CI: 0.740–0.871]; *p* = 0.0001), showing 78% sensitivity and 75% specificity. As a practical metric, HUDmax can be seamlessly incorporated into routine preoperative planning.

Although HUDmax demonstrated strong predictive performance for semirigid access failure, its ability to predict stone-free outcomes was more limited (AUC: 0.697; cutoff: 1.68 mm; sensitivity: 61%, specificity: 71%). This may be partly explained by the pragmatic use of flexible ureteroscopy in both groups. However, while kink-related failure of semirigid URS is an expected outcome, the use of flexible ureteroscopy may not always result in successful treatment. Severe kinking or additional anatomical barriers may still limit the effectiveness of flexible instruments. In our cohort, high HUDmax values continued to compromise stone-free outcomes even after flexible URS was employed. This suggests that HUDmax reflects not only initial access difficulty but also has a significant impact on overall treatment success, as demonstrated by the difference in stone-free rates between groups (45% in SRI vs. 82% in SRA, *p* < 0.0001).

It should also be noted that the lower stone-free rate observed in the SRI group may be attributable to difficulties in maneuverability, impaired visualization, procedural fatigue due to prolonged operative times, and a higher risk of complications in these cases. Accordingly, more frequent placement of double-J stents and the use of percutaneous nephrostomy in selected cases within the SRI group serve as supportive evidence of this procedural burden.

In 2017, Kamo et al. [8] explored the relationship between ureteral kinking and proximal stone location, noting that kinking frequently occurs near the crossing point of the gonadal vein, which may limit spontaneous stone passage. Recent anatomical studies (5, 8) have demonstrated that this region marks a transition where the mobile perirenal ureter becomes fixed to the psoas muscle fascia, predisposing it to dynamic bending during kidney movement. Such severe kinking may locally disturb normal peristalsis and reduce luminal compliance, creating a functional obstruction that can hinder stone passage and complicate endoscopic navigation. CT studies have identified two main stone locations: the proximal ureter and ureterovesical junction, with fewer than 10% located at the ureteropelvic junction or mid-ureter [9,10,11]. Although the exact cause of this distribution remains unclear, factors such as ureteral diameter variations, peristaltic activity, and tortuous anatomy may play a role [12], requiring further investigation. While our study did not evaluate spontaneous passage directly, we found that 74% of SRI cases had maximum kinking within 2 cm distal to the stone compared to only 23% in the SRA group, suggesting that pronounced kinking in this region may cause stones to lodge and complicate endoscopic navigation

The UWT parameter further supports this hypothesis. In our cohort, the median UWT was significantly lower in the SRI group (2.3 mm) than in the SRA group (3.6 mm; *p* = 0.023). This suggests that in kinked segments, the obstruction is more functional and anatomical rather than due to inflammatory wall thickening typically seen with impacted stones. Moreover, HUDmax was frequently located immediately distal to the stone in the SRI group, indicating a possible negative effect on stone passage and procedural success. However, no significant correlation was found between HUDmax and UWT, highlighting the need for further investigation into how kinking interacts with other anatomical and functional parameters.

Our findings showed that, in 26% of cases within the SRI group, the HUDmax point was located more than 2 cm distal to the stone. Notably, 24% of the stones in this group were situated between the sacroiliac joint and the iliac bifurcation. These results suggest that ureteral kinking contributing to URSL failure is not confined to the proximal ureter; it can also occur at other anatomical sites, particularly around the iliac crossing. At this level, external compression and variations in pelvic anatomy may contribute to kinking and adversely affect the outcomes of endoscopic procedures for proximal ureteral stones. Additionally, in our series, no cases of access failure due to kinking were observed among patients with distal ureteral stones. This aligns with previous findings indicating that failed access in this region is typically associated with ureterovesical junction (UVJ) stenosis rather than ureteral kinking [13].

Kinking or tortuosity was observed proximal to the stone in some cases as a secondary obstruction effect. Since this does not impact URSL success, HUDmax was measured distal to the stone. Given the variability of stone locations, HUDmax was classified based on distance to stone rather than anatomical position.

Ureteral kinking is a well-recognized factor that may influence the success of URSL. Zheng et al. [14] reported a 12.2% failure rate in 385 semirigid URS patients, with 4.1% of failures due to anatomical factors like ureteral twists, stenosis, or large angles. Similarly, Turkan et al. [15] found tortuosity and stenosis as common causes of failure in 4.3% of cases. Current preoperative imaging cannot reliably identify anatomical factors related to kinking that may contribute to failures. Our study excluded intrinsic factors (e.g., polyps, strictures, and stone migration) and demonstrated that HUDmax, measured on axial CT images, provides a cost-effective and practical method for assessing kinking.

The CROES study reported median operative times of 40 and 35 min for stones smaller than 2 cm in the proximal and mid-ureter, respectively [3]. Similarly, the SRA group’s median time in our study was 42 min, while the SRI group’s time increased significantly to 55 min. With comparable stone sizes and locations, this finding highlights the adverse impact of ureteral kinking on treatment success and operative time. Additionally, the lack of significant correlations between HUDmax and factors such as stone size, HU, UWT, or hydronephrosis grade suggests that HUDmax may serve as a unique predictor of kinking.

This retrospective matched case–control study has several limitations. Firstly, HUDmax measures only horizontal ureteral displacement, potentially underestimating the severity of kinking in advanced retrograde portions. Secondly, the ureter was not fully visible on NCCT in some cases. In such instances, the maximum observable kinking level was used for the HUDmax calculation, which may have led to an underestimation. Third, although all measurements were carefully standardized and cross-checked internally, no formal analysis of interobserver or intraobserver reproducibility was performed. Future studies should include such testing to confirm the robustness and transferability of HUDmax across different readers. Another limitation is the relatively small sample size, reflecting the rarity of severe ureteral kinking affecting URSL outcomes. In addition, the study was conducted at a single high-volume tertiary center by a limited number of experienced surgeons, which may limit the generalizability of the findings. Therefore, validation through larger, prospective, multicenter studies involving surgeons with diverse levels of experience is necessary. Nevertheless, given the literature’s limited data and the lack of standardized kinking assessment guidelines, we believe our findings, along with HUDmax’s strong predictive value, provide a meaningful contribution.

To our knowledge, this is the first study to simultaneously demonstrate the presence of ureteral kinking on non-contrast CT and its impact on URSL outcomes. These findings underscore the clinical significance of identifying kinking on routine preoperative imaging, as it may inform surgical planning, guide patient counseling, and prompt early preparation for flexible ureteroscopy when necessary. HUDmax, as a quantifiable and reproducible parameter, may serve as a predictive marker for future studies investigating the effect of kinking on retrograde stent placement, ESWL success, and spontaneous stone migration. Even if not widely adopted in clinical practice, HUDmax may offer a standardized and objective measure of kinking severity, contributing meaningfully to scientific research and anatomical classification.

## 5. Conclusions

Although semirigid ureteroscopy failures due to ureteral kinking are rare, their impact on overall treatment outcomes is significant, leading to lower stone-free rates, longer procedures, and increased double-J stent requirements. Therefore, while HUDmax shows promise as a predictive parameter for anatomically challenging cases, its clinical utility should be further verified in larger, multicenter prospective studies before routine implementation.

## Figures and Tables

**Figure 1 medicina-61-01525-f001:**
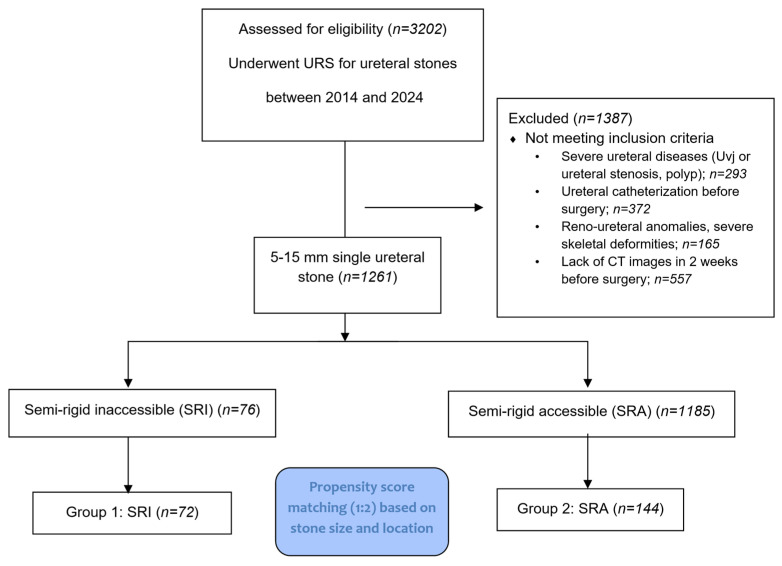
PRISMA flow diagram detailing the process for patient selection criteria.

**Figure 2 medicina-61-01525-f002:**
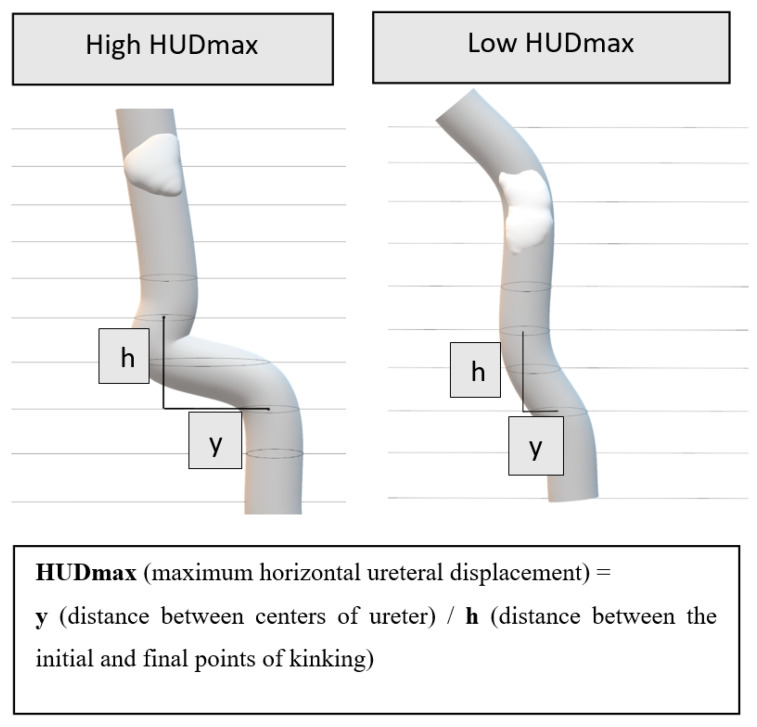
Demonstration of ureteral kinking and HUDmax measurement in 3-D virtual view.

**Figure 3 medicina-61-01525-f003:**
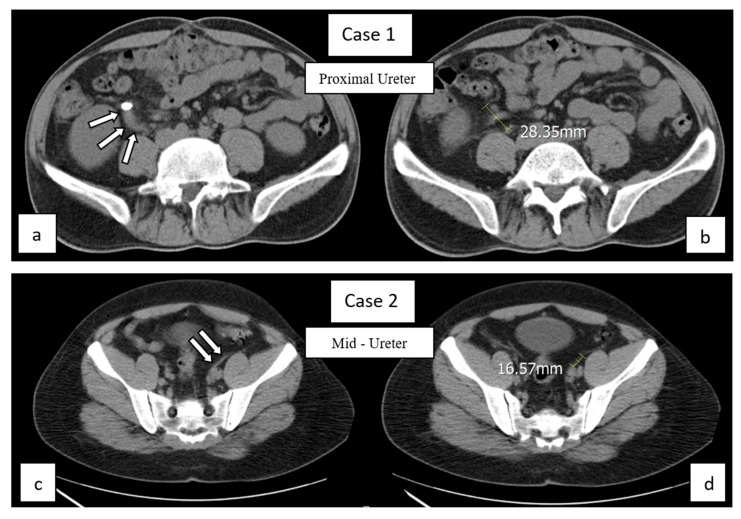
Demonstration of ureteral kinking and HUDmax measurement in real axial NCCT images of two different cases. Arrows and CT images (**a**–**d**) show the measured kinking points and steps, as described in the Measurement of HUDmax section.

**Figure 4 medicina-61-01525-f004:**
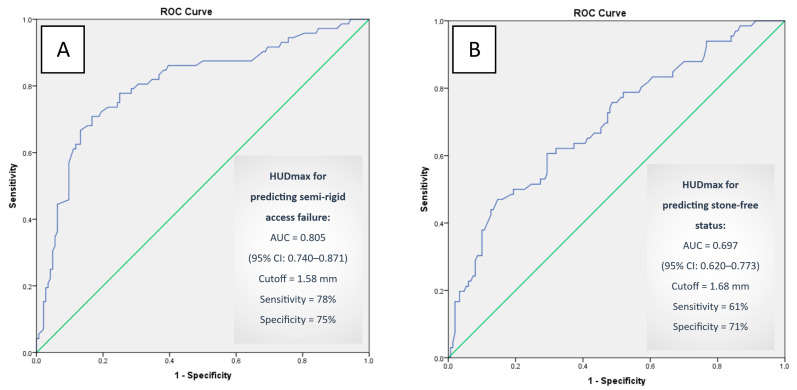
Receiver operating characteristic (ROC) curve analysis demonstrating the predictive value of HUDmax for the access of semirigid ureteroscopy to the stone (**A**) and stone-free status (**B**).

**Table 1 medicina-61-01525-t001:** Patients, stone characteristics, and operative parameters.

	Group 1: SRI *n* = 72	Group 2: SRA *n* = 144	*p* Value
Age (years), med (IQR)	51 (22.8)	48 (23.5)	0.154
BMI (kg/m^2^), mean ± SD	25.8 ± 3.1	25.7 ± 4	0.428
Gender (female/male)	14/58	39/105	0.536
Side (right/left)	35/37	67/77	0.709
Hydronephrosis (grade)			0.106
0–1	32 (44%)	47 (32%)	
2	20 (28%)	51 (36%)	
3–4	20 (28%)	46 (32%)	
Stone area (mm^2^), mean ± SD	65.4 ± 34.9	64.7 ± 30.4	0.874
Stone HU, med (IQR)	960 (596)	831 (486)	0.265
UWT (mm), med (IQR)	2.3 (1.6)	3.6 (2.1)	0.023
Stone location, n(%)			1
Middle ureter	17 (24%)	34 (24%)	
Proximal ureter	55 (76%)	110 (76%)	
HUDmax, med (IQR)	2.36 (1.3)	1.2 (0.8)	0.0001<
HUDmax location			* 0.0001<
Distance to the stone;			
<2 cm, n (%)	53 (74%)	33 (23%)	0.0001<
HUDmax, med (IQR)	2.10 (0.92)	0.96 (1.27)	
>2 cm, n (%)	19 (26%)	111 (77%)	0.0001
HUDmax, med (IQR)	2.04 (1.42)	1.2 (0.60)	
HUD under stone, med (IQR)	2.14 (1.8)	0.42 (0.77)	0.0001<
Duration of surgery (min), med (IQR)	55 (25)	42 (23)	0.0001<
Conversion to Flexible URS	72 (100%)	22 (15%)	0.0001<
Inaccess to stone	72	0	
Stone retropulsion	0	18	
Surgeon preference	0	4	
Placement of safety guidewire at the beginning of the procedure (Proximal to stone)	60 (83%)	124 (86%)	0.735
Stent placement following surgery, n (%)	72 (100%)	120 (83%)	0.0001<
DJ stent	68 (94%)	120 (83%)	
Nephrostomy	4 (6%)	0 (0%)	
Stone free status, n (%)	32 (45%)	118 (82%)	0.0001<

Data are presented as mean ± standard deviation (SD) or median (interquartile range (IQR)) for continuous variables and number of patients (%) for categorical variables. *p* < 0.05 is a significant difference between the groups. SRA: semirigid accessible, SRI: semirigid inaccessible, BMI: body mass index, HUD: horizontal ureteral displacement, HUDmax: maximum horizontal ureteral displacement. * *p*-value for distributions of patients according to HUDmax location.

**Table 2 medicina-61-01525-t002:** Correlation of HUDmax with other parameters.

Parameter	Spearman R	95% Confidence İnterval	*p* Value (Two-Tailed)
Stone size	−0.021	−0.195 to 0.156	0.826
Stone HU	−0.084	−0.267 to 0.107	0.370
UWT	−0.071	−0.240 to 0.118	0.449
Degree of hydronephrosis	0.156	−0.053 to 0.365	0.094
Duration of surgery	0.258	0.089 to 0.433	0.005

## Data Availability

The data that support the findings of this study are not publicly available due to their containing information that could compromise the privacy of research participants, but are available from the corresponding author [UC, e-mail: utkucan99@yahoo.com] upon reasonable request.

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
