# Peer review of "HUDmax as a Novel Parameter in the Assessment of Ureteral Kinking: A Critical Evaluation for Predicting Ureteroscopic Lithotripsy Outcomes"

_medicina, 2025, doi:10.3390/medicina61091525_

Round 1

Reviewer 1 Report

Comments and Suggestions for Authors

A well described study. Abstract is well structured and contains the key points of the study. Introductions is well written and explains the idea of the study. Methods section and statistical analysis are adequate. Results are presented well. Discussion explains the results and contains key citations.  The study proposes a new way to evalutate the difficulty of a ursl procedure using ct scan without contrast. The conclusions of this study are very preliminary and practically need much evaluation in bigger scale studies to accept HUDmax as a predictive tool. A nice study presentation of an idea that need more evalutation.

Author Response

Response to reviewer 1

Thank you very much for taking the time to review this manuscript. Please find the detailed responses below and the corresponding revisions highlighted in the re-submitted files. We sincerely appreciate the reviewer’s constructive comments and positive evaluation, which have greatly helped us improve the clarity and impact of our study.

Reviewer 1

Comment 1:

A well described study. Abstract is well structured and contains the key points of the study. Introduction is well written and explains the idea of the study. Methods section and statistical analysis are adequate. Results are presented well. Discussion explains the results and contains key citations. The study proposes a new way to evaluate the difficulty of a URSL procedure using CT scan without contrast. The conclusions of this study are very preliminary and practically need much evaluation in bigger scale studies to accept HUDmax as a predictive tool. A nice study presentation of an idea that needs more evaluation

Response 1:

We thank the reviewer for this encouraging and positive evaluation of our manuscript. We fully agree that, although HUDmax showed promising results in this retrospective analysis, its clinical applicability should be confirmed by larger, prospective, multicenter studies before widespread adoption. To address this, we have explicitly emphasized this point in the Conclusions section, which now states:

“Therefore, while HUDmax shows promise as a predictive parameter for anatomically challenging cases, its clinical utility should be further verified in larger, multicenter prospective studies before routine implementation.”

We sincerely thank Reviewer 1 for the encouraging and supportive feedback, which helped us further strengthen the clarity and scientific rigor of our manuscript. We hope the revisions meet the expectations for publication.

Reviewer 2 Report

Comments and Suggestions for Authors

Thank you for the opportunity to read this interesting and nicely written manuscript. The authors present a novel parameter - HUDmax (Maximum Horizontal Ureteral Displacement) for quantifying ureteral kinking using non-contrast CT and evaluate its predictive value/usefullness in determining the success of semi-rigid ureteroscopic lithotripsy (URSL). The study is fairly designed, and the central idea is both innovative and clinically relevant. The paper addresses an underexplored anatomical phenomenon that can have a meaningful impact on surgical decision-making and outcomes.

One of the strengths of the study lies in the practical applicability of HUDmax. Because it's derived from standard, non-contrast CT scans, it could be implemented broadly without requiring changes to imaging protocols. The use of propensity score matching for stone size and location is another solid methodological choice, helping reduce potential confounders. The ROC analysis supports the utility of HUDmax as a strong predictor for semi-rigid URSL failure, with an AUC of 0.805, and a reasonable cutoff for clinical use (at least at the point of showing it as a preliminary finding)

However, there are several areas where the manuscript could be strengthened. First, while the retrospective nature of the study is easy to udnersatnd, it introduces limitations around data consistency and possible confounders. In particular, reproducibility of HUDmax measurements could be a concern. The authors do describe a clear measurement protocol, but the manuscript would benefit greatly from reporting interobserver variability or intra-rater reliability data. This would help the reader estimate on how robust and transferable this measurement is across radiologists or urologists with varying levels of experience.

Another area that could use more development is the discussion section. The authors interpret their findings accurately, but a deeper engagement with potential pathophysiological mechanisms behind kinking and its effect on surgical outcomes would be valuable - especially for other practicioners from the different set and settings. For example, is there any literature describing how altered peristalsis or changes in ureteral compliance in kinked regions might reduce stone passage or impair visualization during URSL? Similarly, the lower ureteral wall thickness (UWT) seen in the SRI group is interesting and may suggest functional or developmental differences worth exploring further. Integrating these perspectives could enrich the discussion and frame HUDmax within a broader understanding of ureteral anatomy and physiology.

The authors may also want to consider additional exploratory analyses using their existing dataset. For instance, stratifying HUDmax by stone location (e.g., proximal vs. mid-ureter) or by degree of hydronephrosis could reveal whether kinking plays a variable role depending on regional anatomy. A multivariable logistic regression model that includes HUDmax, UWT, stone size and hydronephrosis might provide a more comprehensive predictive framework. This could help tease out whether HUDmax remains an independent predictor when other known risk factors are accounted for.

Finally, despite the introduction touches on the anatomical challenges of the ureter, it might benefit from a slightly broader context. Ureteral kinking is described primarily as a surgical obstacle, but it might be helpful to mention whether it’s also implicated in spontaneous stone retention or other functional disorders. Framing kinking not just as a surgical nuisance but as a relevant anatomical phenotype may help underline the clinical importance of having a reliable, objective measurement like HUDmax.

To usm it up, I believe this manuscript makes a meaningful contribution to the field of urology and offers a potentially valuable tool for preoperative planning. With some additional clarification on reproducibility, some expansion of the discussion, the manuscript would be further strengthened. I hope the authors will consider these suggestions as they prepare the final version.

Author Response

Response to reviewer 2

Thank you very much for taking the time to review this manuscript. Please find the detailed responses below and the corresponding revisions highlighted in the re-submitted files. We sincerely appreciate the reviewer’s constructive comments and positive evaluation, which have greatly helped us improve the clarity and impact of our study.

Comments 1 (Reproducibility of HUDmax):

First, while the retrospective nature of the study is easy to understand, it introduces limitations around data consistency and possible confounders. In particular, reproducibility of HUDmax measurements could be a concern. The authors do describe a clear measurement protocol, but the manuscript would benefit greatly from reporting interobserver variability or intra-rater reliability data. This would help the reader estimate on how robust and transferable this measurement is across radiologists or urologists with varying levels of experience.

Response 1:

We thank the reviewer for this important observation regarding the reproducibility of HUDmax measurements. While we did not perform a formal interobserver or intraobserver statistical analysis, we took specific steps to ensure measurement consistency and minimize observer bias.

Firstly, all HUDmax measurements were conducted by an experienced endourologist who was familiar with ureteral anatomy and standardized the measurement process using a strict stepwise protocol, as described in the Methods section.

Secondly, representative cases were cross-checked in team meetings with a senior radiologist to verify that the measurement points (horizontal displacement and vertical axis) were consistently selected according to the predefined criteria.

These internal quality control steps enhanced the reliability of the measurements within the scope of this retrospective study. However, we fully agree that formal reproducibility analysis with multiple independent observers is necessary, and we have now acknowledged this as a limitation and a recommendation for future prospective studies.

This has been added to the Discussion, limitatitons section:

“Third, while all measurements were carefully standardized and cross-checked internally, no formal interobserver or intraobserver reproducibility analysis was performed. Future studies should include such testing to confirm the robustness and transferability of HUDmax across different readers.”

Comment 2:

Another area that could use more development is the discussion section. The authors interpret their findings accurately, but a deeper engagement with potential pathophysiological mechanisms behind kinking and its effect on surgical outcomes would be valuable — especially for other practitioners from different settings. For example, is there any literature describing how altered peristalsis or changes in ureteral compliance in kinked regions might reduce stone passage or impair visualization during URSL? Similarly, the lower ureteral wall thickness (UWT) seen in the SRI group is interesting and may suggest functional or developmental differences worth exploring further. Integrating these perspectives could enrich the discussion and frame HUDmax within a broader understanding of ureteral anatomy and physiology.

Response 2:

We thank the reviewer for this suggestion to expand the discussion with more detailed pathophysiological explanations about ureteral kinking and its potential effects on stone passage and URSL outcomes. We agree that providing clearer anatomical and functional context would be beneficial for readers and practitioners from various backgrounds.

Accordingly, we have revised the Discussion section to better integrate the underlying mechanisms of kinking formation and its clinical implications. Specifically, we have added a new consolidated paragraph that combines and clarifies key points about the anatomical transition zone (the crossing point) described by Kamo et al. (2016, 2017, 2021) and its role in altering peristalsis and compliance.

To maintain logical flow and enhance coherence, we repositioned related sentences about the crossing point mechanism, altered peristalsis, functional obstruction, and the significance of UWT immediately after the relevant sections discussing HUDmax and stone-free rates. This reorganization ensures that the anatomical explanation directly supports the clinical findings, providing a stronger link between kinking pathophysiology and our study results.

The newly integrated content can be found in the Discussion, specifically within the fifth, sixth, and seventh paragraphs.

Comment 3:

The authors may also want to consider additional exploratory analyses using their existing dataset. For instance, stratifying HUDmax by stone location (e.g., proximal vs. mid-ureter) or by degree of hydronephrosis could reveal whether kinking plays a variable role depending on regional anatomy. A multivariable logistic regression model that includes HUDmax, UWT, stone size and hydronephrosis might provide a more comprehensive predictive framework. This could help tease out whether HUDmax remains an independent predictor when other known risk factors are accounted for

Response 3:

We greatly appreciate the reviewer’s thoughtful recommendation for additional subgroup analyses and a multivariable logistic regression to further refine the predictive framework.

However, due to our study design — which applied strict inclusion and exclusion criteria and used propensity score matching (PSM) to balance the most critical confounding variables (stone size and location) — the resulting matched sample was relatively small, especially for cases with mid-ureter stones and higher degrees of hydronephrosis. This limited statistical power makes subgroup stratification less reliable within this dataset.

Similarly, while we did explore a preliminary multivariable logistic regression model that included HUDmax, UWT, stone size, and hydronephrosis, the small sample size raised a clear risk of overfitting. Importantly, the PSM itself already controls for major confounders, and the ROC analyses robustly demonstrated the discriminative power of HUDmax as a standalone predictor. Therefore, we believe that adding a regression model would not materially change the interpretation of our results and might even compromise the model’s validity given the limited event count per parameter.

In light of these considerations, we decided not to add further subgroup analyses or a multivariable model to the current manuscript. However, we fully agree with the reviewer that such analyses are methodologically valuable and should be included in future prospective studies with larger cohorts. We have made a note of this point in the limitations section to ensure transparency.

Comment 4:

Finally, despite the introduction touches on the anatomical challenges of the ureter, it might benefit from a slightly broader context. Ureteral kinking is described primarily as a surgical obstacle, but it might be helpful to mention whether it’s also implicated in spontaneous stone retention or other functional disorders. Framing kinking not just as a surgical nuisance but as a relevant anatomical phenotype may help underline the clinical importance of having a reliable, objective measurement like HUDmax.

Response 4:

We fully agree that positioning ureteral kinking within a broader anatomical and functional context could help readers better appreciate its potential implications beyond surgical access difficulty.

However, as this study was specifically designed to evaluate the impact of ureteral kinking on surgical success, and to our knowledge is the first study to quantify kinking for this purpose using non-contrast CT, we did not generate direct evidence about its effect on spontaneous stone passage within this dataset.

We acknowledge that the pathophysiological mechanisms discussed in the Discussion section do suggest a plausible role for kinking in hindering natural stone migration. We have therefore carefully addressed this possibility in the Discussion, highlighting current hypotheses and anatomical explanations based on prior literature.

Additionally, to address this knowledge gap more definitively, we are currently conducting a prospective follow-up study specifically designed to investigate the association between ureteral kinking and spontaneous stone passage outcomes. We believe that the results of that study will provide clearer evidence on this issue in the near future.

In light of this, rather than substantially expanding the Introduction with speculative content, we opted to keep the focus on the established surgical relevance while acknowledging in the Discussion the plausible functional implications and the need for further research, which we believe aligns best with the scope of the current work.

We would like to express our sincere gratitude to Reviewer 2 for their thorough and constructive feedback. The insightful comments and suggestions have significantly strengthened the clarity, methodological rigor, and scientific contribution of our manuscript. We hope that the revisions made fully address all points raised and meet the expectations for publication.

Reviewer 3 Report

Comments and Suggestions for Authors

Introducing HUDmax, a new, readily measured CT-based parameter to evaluate ureteral kinking and predict semi-rigid ureteroscopy (URS) success, this is a successful and clinically significant work. The research is well-organized, with a fair discussion, competent technique, and obvious clinical justification.

Especially in patients undergoing semi-rigid URS, the issue addresses a significant unmet need in endourology: the preoperative prediction of anatomic barriers that may impede URS success. However, it would be appropriate to review a few points before accepting the manuscript:

  • ‘Departmental records of 3,202 patients who underwent semi-rigid ureteroscopy (URS) 73 with or without flexible URS for ureteral stones between 2014 and 2024 at our tertiary care 74 center were retrospectively reviewed.’

The sentence causes conceptual confusion. It has been referred to as semirigid ureteroscopy, but it is unclear which device is used, as it is described as being used with or without a flexible ureteroscope (URS). Please clarify this.

  • When making exclusions in the PLASMA diagram, including the number of patients excluded from each group can strengthen your writing.

  • In cases where resistance was encountered, you said the guidewire was advanced under fluoroscopic guidance. However, was fluoroscopic guidance not used in all your cases, even if you did not encounter any difficulties?

  • Although HUDmax has a good predictive value, prospective investigations are required to validate therapeutic relevance. Emphasize this issue more clealrly in the conclusions. HUDmax should be prospectively confirmed before standard clinical acceptance.

  • The study was conducted at single center, and while not specified specifically, probably with a small number of surgeons. Clearly state this limitation: multi-center validation is required.

  • Flexible URS was used based on "surgeon preference" in some SRA group cases. What does the preference mean?

  • Although the HUDmax calculation is adequately explained, who performed the calculation, inter-observer variability or measurement repeatability is not evaluated.

  • Before publishing, full professional English editing is advised. (Some major language problems like: "we could decide that" → "we conclude that" or "our findings suggest that", "Flexible ureteroscopy were required" → "was required".)

  • The beginning of the discussion section contains sentences very similar to those in the introduction. Consider revising them.

Author Response

Response to reviewer 3

Thank you very much for taking the time to review this manuscript. Please find the detailed responses below and the corresponding revisions highlighted in the re-submitted files. We sincerely appreciate the reviewer’s constructive comments and positive evaluation, which have greatly helped us improve the clarity and impact of our study.

Comment 1:

‘Departmental records of 3,202 patients who underwent semi-rigid ureteroscopy (URS) with or without flexible URS for ureteral stones between 2014 and 2024 at our tertiary care center were retrospectively reviewed.’
The sentence causes conceptual confusion. It has been referred to as semirigid ureteroscopy, but it is unclear which device is used, as it is described as being used with or without a flexible ureteroscope (URS). Please clarify this.

Response 1:  

We thank the reviewer for highlighting this potential source of confusion. We agree that the original phrasing could be clearer. To clarify: all patients initially underwent semi-rigid URS; flexible URS was not systematically used only for failed cases but rather was additionally used when needed, for example if stone position, access angle, or intraoperative findings required flexible instrumentation to complete the procedure effectively.

To prevent misunderstanding, we have rephrased the sentence in the Methods section first paragraph as follows:

“Departmental records of 3,202 patients who underwent semi-rigid ureteroscopy (URS) for ureteral stones, with flexible URS additionally used in selected cases when needed based on intraoperative requirements, between 2014 and 2024 at our tertiary care center, were retrospectively reviewed.”

Comment 2:

When making exclusions in the PLASMA diagram, including the number of patients excluded from each group can strengthen your writing.

Response  2:

We thank the reviewer for this helpful suggestion to enhance the transparency of the patient selection process. In response, we have revised the Figure 1 PLASMA diagram to explicitly include the number of patients excluded at each stage and for each exclusion criterion. This update provides clearer insight into how the final matched cohorts were determined.

The updated flow diagram can be found on Figure 1 in the revised manuscript.

Comment 3:

In cases where resistance was encountered, you said the guidewire was advanced under fluoroscopic guidance. However, was fluoroscopic guidance not used in all your cases, even if you did not encounter any difficulties?

Response 3:

We thank the reviewer for this question. We confirm that fluoroscopic guidance for guidewire placement was used only when necessary — specifically when resistance was encountered or when anatomical conditions required confirmation. In all other routine cases, the guidewire was placed under direct endoscopic vision without additional fluoroscopy.

Comment 4:

Although HUDmax has a good predictive value, prospective investigations are required to validate therapeutic relevance. Emphasize this issue more clearly in the conclusions. HUDmax should be prospectively confirmed before standard clinical acceptance.

Response 4:

We thank the reviewer for emphasizing this important aspect. In line with this recommendation, we have now explicitly added this point to the Conclusions section to clearly state that HUDmax should be validated by larger, prospective, multicenter studies before being adopted in routine clinical practice.

The revised sentence reads:

“Therefore, while HUDmax shows promise as a predictive parameter for anatomically challenging cases, its clinical utility should be further verified in larger, multicenter prospective studies before routine implementation.”

Comment 5:

The study was conducted at single center, and while not specified specifically, probably with a small number of surgeons. Clearly state this limitation: multi-center validation is required.

Response 5:

We thank the reviewer for highlighting this important point. We agree that the single-center design and limited number of surgeons could affect the generalizability of our findings.

To address this, we have connected this limitation directly to the existing sentence in the Limitations section (Discussion) to maintain clear flow. The revised part now reads:

“ In addition, the study was conducted at a single high-volume tertiary center by a limited number of experienced surgeons, which may limit the generalizability of the findings. Therefore, validation through larger, prospective, multicenter studies involving diverse surgeon populations is necessary.”

Comment 6:

Flexible URS was used based on "surgeon preference" in some SRA group cases. What does the preference mean?

Response 6:

We thank the reviewer for requesting clarification. We have now integrated a more precise explanation directly into the existing sentence in the Methods/Ureteroscopy procedure section, 2nd paragraph, which now reads:

“Flexible ureteroscopy was required in all cases where the stone could not be reached with a semi-rigid ureteroscope, in cases of stone retropulsion into the kidney, or when, based on the surgeon’s intraoperative judgment, flexible URS was considered preferable to optimize maneuverability, access difficult angles, or minimize ureteral trauma even if semi-rigid access was technically possible.”

This clarifies exactly what “surgeon preference” means in our practice.

Comment 7:

Although the HUDmax calculation is adequately explained, who performed the calculation, inter-observer variability or measurement repeatability is not evaluated.

Response 7:

We thank the reviewer for highlighting this important methodological point. All HUDmax measurements were performed by a single experienced endourologist using a standardized stepwise protocol as described in the Methods section. Additionally, representative cases were cross-checked with a senior radiologist to maintain measurement consistency.

However, a formal inter-observer or intra-observer reliability analysis was not performed due to the retrospective design. This limitation is now clearly acknowledged in the Limitations section (Discussion), and we have recommended that future prospective studies include reproducibility testing with multiple independent readers to confirm the robustness and transferability of HUDmax

Comment 8:

Before publishing, full professional English editing is advised. (Some major language problems like: "we could decide that" → "we conclude that" or "Flexible ureteroscopy were required" → "was required".)

Response 8:

We sincerely thank the reviewer for pointing out these language issues. In response, the entire manuscript has undergone a careful, line-by-line language revision to correct grammatical errors and improve academic phrasing. Key examples, such as verb agreement and word choice, have been corrected as suggested (e.g., “Flexible ureteroscopy was required”). All changes are highlighted in the revised version.

Comment 9:

The beginning of the discussion section contains sentences very similar to those in the introduction. Consider revising them.

Response 9:

We thank the reviewer for this valuable observation. We fully agree that unnecessary repetition should be avoided to improve the clarity and focus of the Discussion. Accordingly, we have completely revised the opening paragraphs of the Discussion section. The new introduction now directly emphasizes the novelty and clinical relevance of HUDmax as a practical, non-contrast CT-based parameter for predicting URSL outcomes, and clearly distinguishes this section from the Introduction. The updated first and second paragraphs can be found in the revised manuscript (Discussion, paragraphs 1–2).

We sincerely thank Reviewer 3 for the thoughtful and constructive feedback. The detailed comments and suggestions have greatly contributed to improving the clarity, methodological rigor, and scientific value of our manuscript. We hope that the revisions fully address all points raised and meet the journal’s standards for publication.

Reviewer 4 Report

Comments and Suggestions for Authors

This study investigates HUDmax, a novel parameter as a predictor of ureteral kinking, and its implications in URS. The manuscript is well-structured and methodologically sound. The concept is innovative and addresses a gap in the existing literature. The aim of this study is to predict access failure and surgical complexity in URS. The authors presented a well-designed, statistically rigorous manuscript in which the outcome measures are clearly defined. 

In my opinion, there is just one concern that has to be addressed: the measurement method is well-described, and the illustrations help with its interpretation; however, the authors should address the reproducibility and interobserver reliability in the discussion, and if not addressed, it should be mentioned as a limitation of the study.

Other than this, the manuscript is suitable for publication as soon as this is addressed. 

Author Response

Response to reviewer 4

Thank you very much for taking the time to review this manuscript. Please find the detailed responses below and the corresponding revisions highlighted in the re-submitted files. We sincerely appreciate the reviewer’s constructive comments and positive evaluation, which have greatly helped us improve the clarity and impact of our study.

Comments 1:

The measurement method is well-described, and the illustrations help with its interpretation; however, the authors should address the reproducibility and interobserver reliability in the discussion, and if not addressed, it should be mentioned as a limitation of the study.

Response 1:  

We thank the reviewer for highlighting this important point. We fully agree that the reproducibility and interobserver reliability of HUDmax are key considerations for its clinical adoption. As described in the Methods section, all measurements were performed by a single experienced endourologist using a standardized stepwise protocol, with representative cases cross-checked by a senior radiologist to ensure consistency. However, a formal interobserver or intraobserver reproducibility analysis was not conducted due to the retrospective design.

To address this concern, we have explicitly acknowledged this as a limitation in the Limitations section of the Discussion and recommended that future prospective studies should include reproducibility testing across multiple independent readers to validate the robustness and generalizability of HUDmax.

We sincerely thank Reviewer 4 for the supportive and encouraging feedback. We appreciate the insightful comment regarding the reproducibility of HUDmax, which we have now fully addressed as recommended. We hope that the revised manuscript meets the reviewer’s expectations and is now suitable for publication.

Round 2

Reviewer 2 Report

Comments and Suggestions for Authors

As a reviewer, I’m pleased to see that the authors have thoroughly addressed the comments with thoughtful and transparent responses. The revisions meaningfully strengthen the manuscript’s clarity, contextual depth and methodological rigor. I especially appreciate the clear acknowledgment of limitations and the well-justified decisions regarding additional analyses.

At this point I don't have any further comments and I am supportive of a publication of the paper.